# Preiswerkite: A First Occurrence in Marble Hosting Gem Spinel Deposits, Luc Yen, Vietnam

Vladimir G. Krivovichev [1],*[ID], Katherine A. Kuksa [2], Pavel B. Sokolov [3], Olga Yu. Marakhovskaya [1],
Andrey A. Zolotarev [4][ID], Vladimir N. Bocharov [5][ID], Tatyana F. Semenova [4], Maria E. Klimacheva [1]
and Geir Atle Gussiås [6]

[1] Department of Mineralogy, Institute of Earth Sciences, St. Petersburg State University, University Emb. 7/9, 199034 St. Petersburg, Russia
[2] Department of Geomorphology, Institute of Earth Sciences, St. Petersburg State University, University Emb. 7/9, 199034 St. Petersburg, Russia
[3] SOKOLOV Co., Ltd., Gatchinskaya Str., 11/A, 7N, 197136 St. Petersburg, Russia
[4] Department of Crystalography, Institute of Earth Sciences, St. Petersburg State University, University Emb. 7/9, 199034 St. Petersburg, Russia
[5] Geo Environmental Centre "Geomodel", The Research Park, St. Petersburg State University, Ul'yanovskaya Str. 1, 198504 St. Petersburg, Russia
[6] BalderGems Co., Yen The YB 36000, Vietnam
* Correspondence: v.krivovichev@spbu.ru

**Abstract:** We report a new occurrence of preiswerkite, the rare sodium analog of eastonite, the trioctahedral mica, from marble-hosted noble spinel deposits of the Luc Yen district, northern Vietnam. It is found in marble for the first time. The preiswerkite is anhedral and associated with phlogopite, aspidolite, sadanagaite, pargasite, spinel, corundum, dolomite and calcite. The average compositions of preiswerkite is $(Na_{0.88}Ca_{0.08}K_{0.01})_{\Sigma 0.97}(Mg_{2.29}Al_{0.72}Fe_{0.04})_{\Sigma 3.05}[(Al_{1.95}Si_{2.05})_{\Sigma 4.00}O_{10}](OH)_2$. The compositions of preiswerkite have a narrow range of Mg# values (0.96–0.99) and define a preiswerkite-aspidolite solid-solution series. Compared with other occurrences, the Luc Yen preiswerkite has a low iron content, which attains 0.09 atoms per formula unit (1.53 wt.% FeO). The formation of preiswerkite is favored by the proportion of Mg, Al and Si in the precursor rocks and the increased activity of sodium and $H_2O$ in the fluid phase.

**Keywords:** preiswerkite; aspidolite; trioctahedral mica; Raman spectrum; noble spinel; marble-hosted deposits; Luc Yen district; Vietnam

## 1. Introduction

The Luc Yen deposit of Vietnamese cobalt blue spinel is located in northern Vietnam in Yen Bai province. Discovered at the end of the last century [1] and then distinguished as an important source of gem-quality spinel and corundum [2–4], this deposit immediately attracted the close attention of researchers. Some issues of spinel composition of the Luc Yen deposit have been discussed [3,5–7], mainly to identify diagnostic features that allow distinguishing spinel entering the jewelry market from different deposits of the world. Special studies concern the nature of the mineral coloration [1,8], assessment of the age of the host marbles [9] and the conditions of their formation [10–15]. The greatest attention is paid to the geology of spinel-containing marbles in Ref. [3], where for the first time the assumption was made about the different genesis of red and purple spinel, formed as a result of isochemical metamorphism and metasomatism, respectively. However, little attention was paid to the impact of metasomatic processes on the spinel-bearing marbles at Luc Yen.

Aspects of the detailed mineralogy of the spinel-bearing marbles [15] indicate some involvement of hydrothermal fluids into the genesis of spinel. The aim of the current

work is to present detailed information on the first occurrence of the rare Na-Al-rich mica, preiswerkite, in the Luc Yen ruby and gem spinel marble-hosted deposit of northern Vietnam (the first mineralogical data for this genetic type). We describe its chemical composition, Raman spectroscopy and X-ray data and discuss its significance to marble genesis.

Preiswerkite, ideally $Na(Mg_2{}^{VI}Al)_{\Sigma 3}({}^{IV}Al_2Si_2O_{10})(OH)_2$, is the more aluminous end member of the trioctahedral (K, Na)-Mg-micas (Figure 1). Since its discovery in 1980 [16], this rare mineral has been reported in only ten other locations: Allalinhorn, Switzerland [17,18]; Liset and Blengsvatn, Norway [19,20]; La Copointrie, Vedée, France [21]; Koralpe and Saualpe, Austria [22]; Rio La Palmilla, Guatemala [23]; the Dabie UHP metamorphic belt, China [24,25]; the Aktyuz eclogite, Kyrgyzstan [26]; the Kechros metamorphic complex, Greece [27]; the Khoy ophiolite in Iran [28]; the Vumba schist belt, Botswana [29]. These studies have shown that preiswerkite typically occurs in metabasic and meta-ultrabasic rocks in which it formed during a retrograde metamorphism of the eclogite or amphibolite facies [16–21]; rarely, it is found in meta-acidic rocks during a prograde stage metamorphism to the greenschist to amphibolite facies [20].

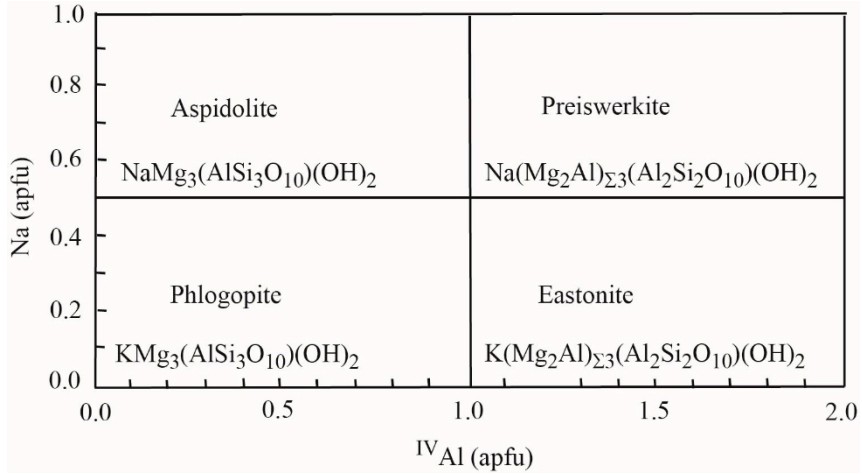

**Figure 1.** Classification diagram of trioctahedral (K,Na)-Mg micas.

## 2. Geological Setting and Petrography

The Luc Yen ruby and gem spinel deposit is located in the Lo Gam metamorphic zone, which, together with the Day Nui Con Voi Range, constitutes part of the Red River shear zone of northern Vietnam [9]. The Lo Gam zone consists of thick Cambrian marble units overlying micaschists [11]. Besides carbonates (calcite, dolomite), the marble units around the Luc Yen deposit contain forsterite, pargasite, sadanagaite, phlogopite, aspidolite, pyrite, pyrrhotite, graphite and dravite. Thus, these minerals indicate that mineral formation in the Luc Yen district occurred not only in amphibolite but also in granulite facies, as previously indicated by Garnier et al. [10].

Five color varieties of spinel have been found in the Luc Yen marbles: (1) lavender and purplish in marbles with symplectic texture together with calcite, dolomite, forsterite, pargasite, clinohumite and rare graphite, (2) light to vivid red with calcite, dolomite, forsterite and green pargasite, in some cases overgrowing spinel grains, (3) purplish violet to brownish spinel intergrown with corundum in marbles with symplectitic texture together with pyrrhotite, pyrite and titanite, (4) vivid-blue spinel in calcite marbles with chondrodite, rare phlogopite and abundant graphite, and (5) blue with calcite, dolomite, forsterite and grayish-green pargasite [15].

Preiswerkite aggregates have been observed in marbles in association with type 3 violet to brownish spinel overgrowing corundum grains. It consists of pale brown to brown elongated prismatic aggregates up to 0.5 mm in length (rarely up to 2 mm), with a perfect cleavage typical of a mica. Preiswerkite associates with amphibole (sadanagaite), phlogopite, aspidolite, chlorite, spinel, corundum, calcite, rutile, pyrite and pyrrhotite. It

mostly occurs as a pseudomorph after phlogopite (or aspidolite), rarely as inclusions in spinel and is altered partially to chlorite at the rims (Figure 2).

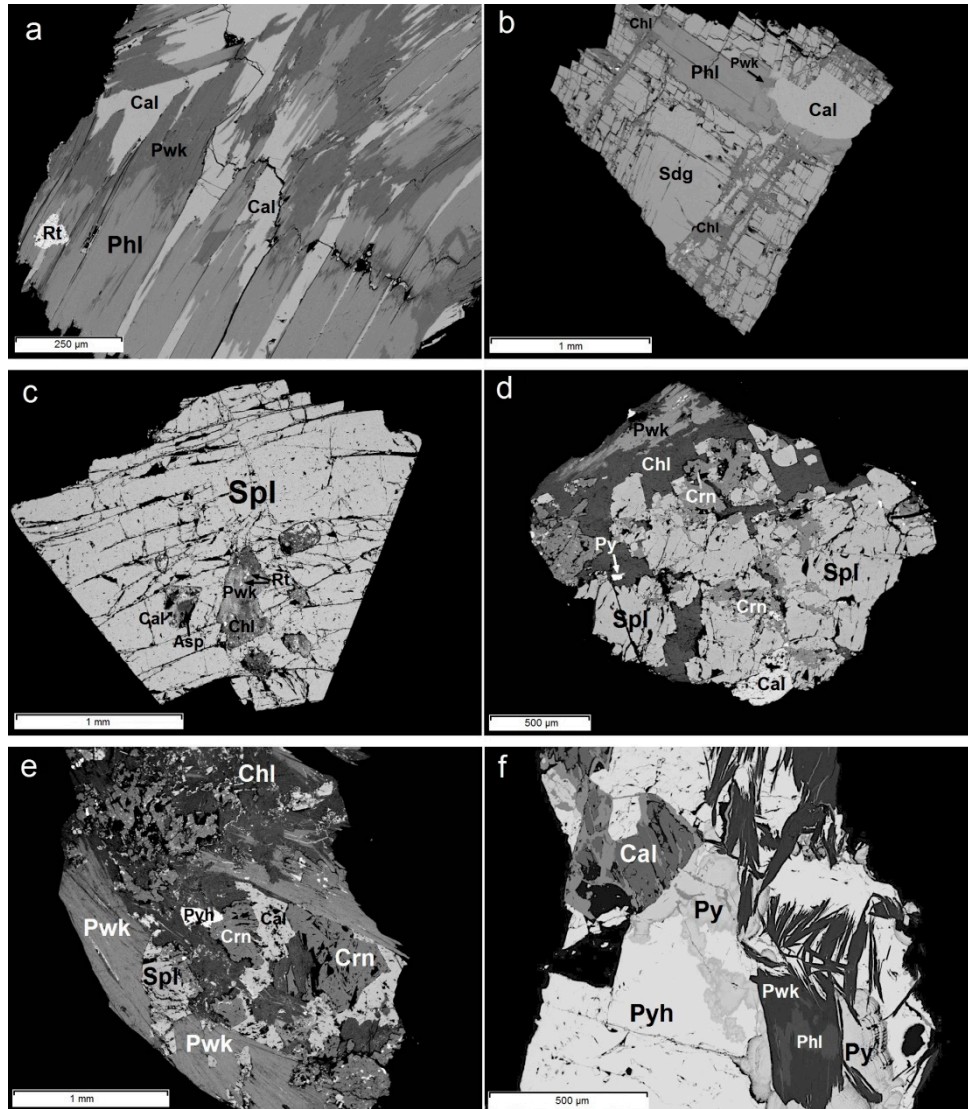

**Figure 2.** Back-scattered electron scanning electron microscopy images of a polished section of the Luc Yen preiswerkite: (**a**) preiswerkite replacing phlogopite in calcite matrix; (**b**) preiswerkite replacing phlogopite in association with sadanagaite and calcite; (**c**) preiswerkite + chlorite + rutile inclusions in spinel grains; (**d**) preiswerkite altered to chlorite at the rim in association with violet spinel replacing corundum; (**e**) preiswerkite and chlorite in association with violet spinel overgrowing corundum grains; (**f**) preiswerkite replacing phlogopite in association with calcite, pyrite and pyrrhotite. Mineral symbols [30]: Asp—aspidolite; Cal—calcite; Chl—chlorite; Crn—corundum; Phl—phlogopite; Pwk—preiswerkite; Py—pyrite; Pyh—pyrrhotite; Rt—rutile; Sdg—sadanagaite; Spl—spinel.

## 3. Methods

Major element compositions of preiswerkite from thin sections of samples were determined at the Electron Microprobe Laboratory in the Geomodel Center at St. Petersburg University using a JEOL-8200 Electron Microprobe. Natural standards were used for calibration. An operating voltage of 15 kV and a beam current of 20 nA were used with a beam diameter of 1 μm (5 μm spot size). Representative results of the electron-microprobe analyses of the Luc Yen preiswerkite are listed in Table 1.

**Table 1.** Results of electron-microprobe analyses of preiswerkite from the Luc Yen deposits.

|  | 1137a | 1144 | 1038 | 323 | 164 | 1149 | $\overline{X}$ | σ |
|---|---|---|---|---|---|---|---|---|
| $SiO_2$ wt.% | 29.35 | 29.96 | 31.21 | 31.91 | 31.74 | 29.90 | 30.68 | 1.08 |
| $Al_2O_3$ | 33.22 | 33.04 | 32.52 | 31.14 | 31.46 | 32.81 | 32.36 | 0.86 |
| * FeO | 0.47 | 0.46 | 1.53 | 0.95 | 0.92 | 0.46 | 0.80 | 0.43 |
| MgO | 22.95 | 23.44 | 21.22 | 22.29 | 22.25 | 22.67 | 22.47 | 0.75 |
| CaO | 1.41 | 1.45 | 0.20 | 0 | 0.26 | 1.39 | 0.78 | 0.70 |
| $Na_2O$ | 6.58 | 6.52 | 7.23 | 7.51 | 7.29 | 6.50 | 6.94 | 0.45 |
| $K_2O$ | 0.49 | 0.83 | 1.14 | 1.18 | 1.08 | 0.49 | 0.87 | 0.32 |
| ** $H_2O$ | 4.42 | 4.27 | 4.38 | 4.42 | 4.42 | 4.40 | 4.38 | 0.06 |
| Total | 98.89 | 100.09 | 99.43 | 99.4 | 99.42 | 98.62 | 99.31 | 0.51 |
| *** Si *apfu* | 2.01 | 2.01 | 2.13 | 2.16 | 2.15 | 2.05 | 2.08 | 0.07 |
| $^{IV}$Al | 1.99 | 1.99 | 1.87 | 1.84 | 1.85 | 1.95 | 1.91 | 0.07 |
| Total | 4.00 | 4.00 | 4.00 | 4.00 | 4.00 | 4.00 | 4.00 | 0 |
| $^{VI}$Al | 0.65 | 0.63 | 0.75 | 0.65 | 0.67 | 0.77 | 0.69 | 0.06 |
| Fe | 0.03 | 0.03 | 0.09 | 0.05 | 0.05 | 0.01 | 0.04 | 0.03 |
| Mg | 2.32 | 2.35 | 2.16 | 2.25 | 2.26 | 2.35 | 2.28 | 0.07 |
| Total | 3.00 | 3.01 | 3.00 | 2.95 | 2.98 | 3.13 | 3.01 | 0.06 |
| Ca | 0.10 | 0.10 | 0.01 | 0.00 | 0.02 | 0.06 | 0.05 | 0.04 |
| Na | 0.87 | 0.85 | 0.96 | 0.99 | 0.96 | 0.77 | 0.90 | 0.08 |
| K | 0.02 | 0.03 | 0.05 | 0.05 | 0.05 | 0.04 | 0.04 | 0.01 |
| Total | 0.99 | 0.98 | 1.02 | 1.04 | 1.03 | 0.87 | 0.99 | 0.06 |
| Mg# | 0.99 | 0.99 | 0.96 | 0.98 | 0.98 | 0.99 | 0.98 | 0.01 |

* Total Fe as FeO; ** calculated from stoichiometry; *** atomic proportions based on 11 O; Mg#—Mg/(Mg+Fe); $\overline{X}$—arithmetic mean; σ—standard deviation.

X-ray powder diffraction data of preiswerkite were obtained using a diffractometer Rigaku R-Axis Rapid II (Gandolfi method, Debye–Scherrer geometry, d = 127.4 mm) equipped with a rotating anode X-ray source (CoKα, λ = 1.79021 Å) and a curved image plate detector at X-ray Diffraction Centre of St. Petersburg University. The data were integrated using the software package Osc2Xrd [31]. Unit-cell parameters were refined using PDXL2 software [32].

The Raman spectrum of preiswerkite was recorded by means of Horiba Jobin-Yvon LabRam HR800 spectrometer using solid-state laser with λ = 532 nm (power on the sample 15 mW) and 40× objective. The sample was oriented randomly and measured at room temperature. The data were obtained in the range of 70–4000 cm$^{-1}$ and 2 cm$^{-1}$ spectral resolution. The calibration was carried out using Si standard (520.7 cm$^{-1}$).

## 4. Results

### 4.1. Major Element Mineral Chemistry

The $SiO_2$, $Na_2O$, $Al_2O_3$ and MgO contents of the Luc Yen preiswerkite are similar to those reported in previous studies [19–27]. The microprobe analysis gives an averaged structural formula of $(Na_{0.88}Ca_{0.08}K_{0.01})_{\Sigma0.97}(Mg_{2.29}{}^{VI}Al_{0.72}Fe_{0.04})_{\Sigma3.05}[(^{IV}Al_{1.95}Si_{2.05})_{\Sigma4.00}O_{10}]$ $(OH)_2$, which is close to the ideal formula, $Na(Mg_2{}^{VI}Al)_{\Sigma3.00}[(^{IV}Al_2Si_2)_{\Sigma4.00}O_{10}](OH)_2$. The Mg# is higher in the Luc Yen preiswerkite than in others reported to date. The Luc Yen preiswerkite is characterized by small amounts of CaO (0 to 1.45 wt.%) and $K_2O$ (0.49 to 1.18 wt.%; Table 1). A characteristic geochemical feature of our preiswerkite is a low iron content, which reaches 0.09 atoms per formula unit (1.53 wt.% FeO). The contents of Ti, F and Cl are below the detection limits.

### 4.2. X-ray Powder-Diffraction Data

Too little material was available (only one very small crystal) for powder diffraction analysis. Consequently, a quasi-random powder dataset was obtained using the Gandolfi

method. The crystal was mounted on a Rigaku R-Axis Rapid II curved imaging plate microdiffractometer and a dataset collected using Co$K\alpha$ radiation. A Gandolfi-type randomized crystal movement was achieved by rotations on the $\varphi$ axis. Unit cell parameters refined from this experiment are given in Table 2. These data are in agreement with those of Oberti et al. [33], who present single crystal X-ray structure refinements of preiswerkite from the type locality [16]. Observed *d* values and intensities were obtained by profile fitting using the PDXL2 software [32].

**Table 2.** Unit cell parameters of preiswerkite.

|  | 1 | 2 | 3 | 4 | 5 |
|---|---|---|---|---|---|
| *a* (Ǎ) | 5.239 (2) | 5.2392 (18) | 5.225 (4) | 5.228 (7) | 5.243 (2) |
| *b* (Å) | 9.056 (3) | 9.057 (3) | 9.050 (8) | 9.049 (10) | 9.055 (3) |
| *c* (Å) | 9.791 (3) | 9.792 (3) | 9.791 (9) | 9.819 (12) | 9.793 (3) |
| $\beta$ (º) | 100.201 (15) | 100.204 (14) | 100.27 (6) | 100.41 (13) | 100.264 (16) |
| V (Å$^3$) | 457.09 | 457.21 | 455.6 | 456.9 | 457.5 (3) |

1, 2—Luc Yen (our data); 3, 4—Oberti et al. [33]; 5—DB card number 00-042-0605.

*4.3. Raman Spectroscopy*

The Raman spectrum of preiswerkite from the Luc Yen deposit is very similar to the spectra of preiswerkite from Khoy, Iran [28] and Liset, Norway [19], although the Luc Yen preiswerkite spectrum is of higher quality in terms of signal/noise.

The following main bands were observed in the Raman spectrum of preiswerkite from the Luc Yen deposit (strong bands are marked in bold, cm$^{-1}$): **112**, **220**, **294**, 314, 380, 448, 490, 550, **650**, 811, 913, 958 and **3702** cm$^{-1}$ (Figure 3). Peaks positions are defined by the mineral structure, mainly due to interrelations between Si, Al and O [34]. Alternating tetrahedra in the preiswerkite structure consist of the Al and Si cations, respectively, bonded with three basal O atoms. In addition, Si-O$_b$-Al bonds as well as Si-O$_a$ vibrations where Si is surrounded by the three tetrahedra (AlO$_4$) and Al-O$_a$ vibrations where Al is surrounded by the three tetrahedra (SiO$_4$) (here subscript "a" indicates apical atom and "b" is basal atom) are important structure elements, defining Raman spectral features [33].

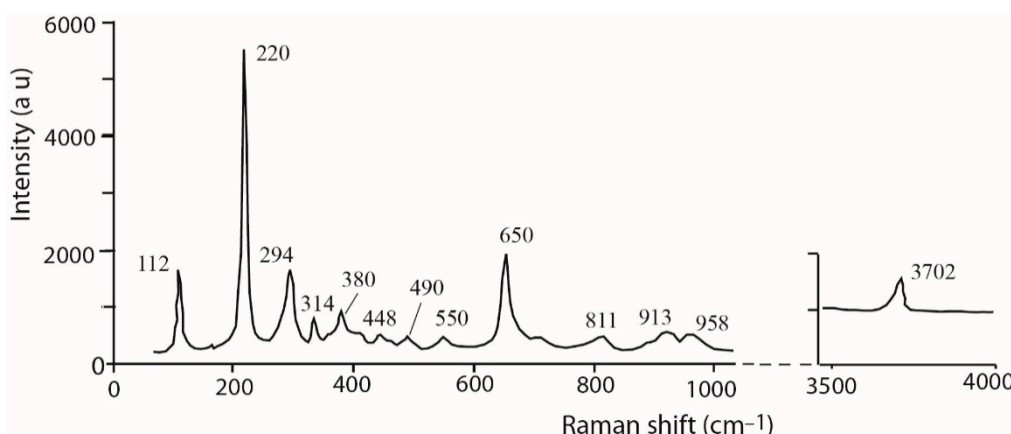

**Figure 3.** Raman spectra for the Luc Yen preiswerkite specimen.

The bands in the Raman spectrum are assigned according to Raman spectra reported in previous studies (e.g., [28,34–36]). The most intensive band at 650 cm$^{-1}$ can be assigned to the Si-O$_b$-Al vibration. Among all trioctahedral micas, preiswerkite has the highest Al content, thus the intensity of this band in the preiswerkite Raman spectrum reaches a maximum. At the same time, the intensity of the tetrahedron vibration decreases and becomes very weak. The profile of the band at 652 cm$^{-1}$ has asymmetry to high wavenumbers and, according to [34], overwhelms the two weak bands at 679 cm$^{-1}$ (Si-O-Si) and 655 cm$^{-1}$ (Al-O-Al). Bands at 958 cm$^{-1}$ and 913 cm$^{-1}$ can be attributed to Si-O$_a$ vibration

and Al-O$_a$ vibration, respectively [33]. The strongest spectrum band at 220 cm$^{-1}$ is assigned to asymmetrical stretching of the isosceles triangle O-H-O composed of the proton and two adjacent apical oxygen atoms. The band at 294 cm$^{-1}$ is attributed to symmetrical stretching of the isosceles triangle O-H-O [35,36]. The distinct Raman band at 3702 cm$^{-1}$ is related to OH-stretching vibration modes [34]. The other observed bands can be attributed to the bending vibration of the tetrahedra, M-O6 vibrations and lattice modes.

## 5. Discussion

### 5.1. Geochemical and Crystallochemical Consideration

The rarity of preiswerkite in nature is believed to be mainly due to the physicochemical parameters of the mineral-forming medium, the behavior of K and Na in geochemical processes and the crystallographic configuration of mica-group minerals, rather than unusual P-*T* conditions [21]. This section is devoted to the precise crystallographic constraints that may favor preiswerkite formation.

Compositionally, the total population of ~6750 analyses of micas is subdivided by Tischendorf et al. [37] into two groups: (1) true micas, with monovalent cations in the interlayer (96.8% of all analyses), and (2) brittle micas containing divalent cations in the interlayer (3.2%). In turn, depending on the interlayer cations, true micas were divided into two subgroups: (1a) true K micas (94.7% of all analyses) (phlogopite, annite, muscovite, etc.), and (1b) true non-K micas (2.1%) containing the monovalent cations Na, Rb, Cs or NH$_4$ as species-defining elements substituting for K (aspidolite, preiswerkite, paragonite, etc.). Replacement of K by Na in phlogopite and eastonite (Na > K, a.p.f.u.) leads to aspidolite and preiswerkite, respectively. Thus, trioctahedral true potassic micas are more common in nature than true sodic micas (preiswerkite, aspidolite, ephesite and wonesite), which are found only in small quantities and in some localities.

This cannot be explained from a geochemical point of view by the contents of K and Na in the earth's crust. Thus, according to Christy [38] the elemental abundance by weight % of K and Na are approximately the same (20.9 and 23.6 wt.%), and the elemental abundance by number of atoms of K is about two times smaller than for Na (11,100 and 21,300 p.p.m, respectively) [38].

The equilibrium between preiswerkite and eastonite can be represented as the following exchange reaction:

$$Na(Mg_2Al)(Al_2Si_2O_{10})(OH)_2 + K^+ = K(Mg_2Al)(Al_2Si_2O_{10})(OH)_2 + Na^+$$

$$\text{Preiswerkite} \qquad\qquad\qquad \text{Eastonite} \qquad\qquad\qquad (1)$$

The equilibrium conditions of any reactions in which the solutions take part are determined by the expression:

$$\Delta_r G_{T,p} = 0 = \Delta_r G_{T,p=1} + \Delta_r V_{\text{solid}} P + RTlnK \qquad (2)$$

where $\Delta_r G_{T,p=1}$ *and* $\Delta_r G_{T,p}$ are the isobaric potentials of the reaction between components in standard state at given *T* and pressure (1 and P bar, respectively); *K* is the equilibrium constant of the reaction.

Unfortunately, thermodynamic data of preiswerkite (and aspidolite) are not available in the literature. Therefore, for an approximate estimation of the Na/K ratio in a mineral-forming medium in equilibrium with K-Na micas, we used the equilibria between muscovite and paragonite, for which thermodynamic data are available.

$$NaAl_2(AlSi_3O_{10})(OH)_2 + K^+ = KAl_2(AlSi_3O_{10})(OH)_2 + Na^+ \qquad (3)$$

$$\text{Paragonite} \qquad\qquad \text{Muscovite}$$

Let us calculate equilibrium constant for the last reaction:

$$lgK = lg(a_{Na^+}/a_{K^+}) = -(\Delta_r G_{T,p=0.1} + \Delta_r V_{solid}P)/2.3026RT. \quad (4)$$

Obtaining thermodynamic data from [39], we obtain for 300 °C and 3 kbar:

$$lg(a_{Na^+}/a_{K^+}) = 3.02; \text{ and } a_{Na^+}/a_{K^+} = 10^{3.02} = 1054. \quad (5)$$

It follows that paragonite formation requires the activity (concentration) Na$^+$ in solution to be about three orders of magnitude higher than the activity of K$^+$. The main reason for this is the crystal chemical features of the mica group minerals: the interlayer position in the mica structure is "more comfortable" for a larger K$^+$ compared with sodium Na$^+$ ($R_i$ (ionic radii) = 1.33 Å and 0.98 Å, respectively). Thus, sodic micas are inherently less likely to form because of the relatively small ionic radius of the interlayer cation and the stronger repulsion due to the closeness of the juxtaposed sheets of tetrahedra that results. As a corollary statement, eastonite is more likely to form than preiswerkite.

It should be noted that mica-group minerals are commonly formed during retrograde metamorphism as a result of amphibole substitution. Similar to micas, amphiboles (theoretical formula: $AB_2C_5(T_8O_{22})X_2$) may contain from zero to one atom or ion per formula in A site. But in most amphiboles, the A site has CN X rather than XII, because there are two sets of five oxygen around each A site cation, rather than two sets of six in micas. Na$^+$ in A sites electrically compensates charge deficiency induced by Si$^{4+}$ ↔ Al$^{3+}$ exchange: $\square$ + [Si$^{4+}$] ↔ Na$^+$ + [Al$^{3+}$] (solid-solution series tremolite–edenite, Figure 4). In solid-solution series edenite–pargasite–sadanagaite, the heterovalent substitution and consequent electric charge balance is effected as follows: Mg$^{2+}$ + [Si$^{4+}$] ↔ Al$^{3+}$ + [Al$^{3+}$] (Figure 4). The same type of substitution occurs in solid-solution series phlogopite–eastonite for potassic micas and aspidolite–preiswerkite for sodic micas (Figure 4).

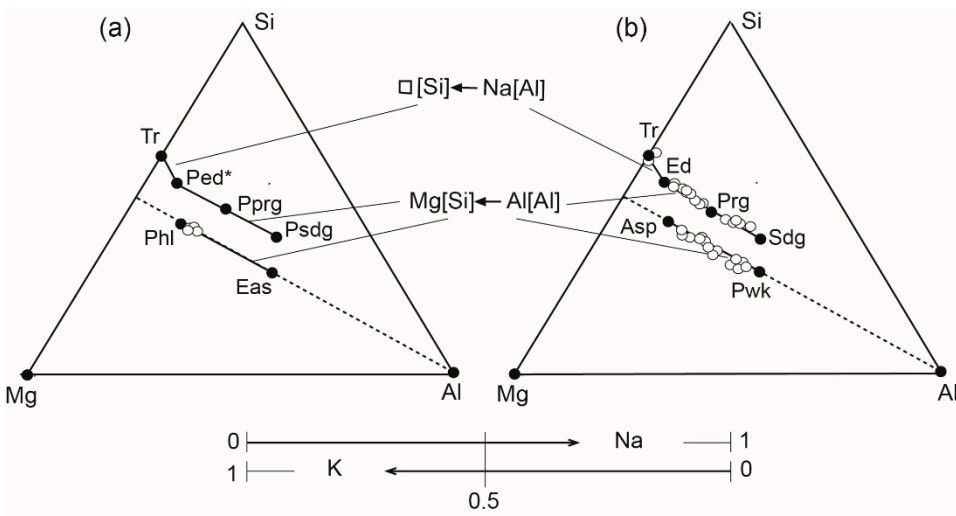

**Figure 4.** Chemographic representation of trioctahedral mica and calcium amphibole compositions. Mica compositions presented in the following lines: phlogopite (Phl), eastonite (Eas) for potassic micas (**a**) and aspidolite (Asp), preiswerkite (Pwk) for sodic micas (**b**). Amphibole compositions are presented in the following lines: tremolite (Tr), potassic–edenite (Ped*, theoretical), potassic–pargasite (Pprg), potassic–sadanagaite (Psdg) for potassic amphiboles (a) and tremolite (Tr), edenite (Ed), pargasite (Prg), sadanagaite (Sdg) for sodic amphiboles (b). Empty circles indicate the compositions of the Luc Yen micas and amphiboles.

## 5.2. Analysis of Mineral Parageneses

The main minerals, which reflect the most essential associations with preiswerkite are phlogopite, aspidolite, pargasite, sadanagaite, corundum, spinel, anorthite, dolomite

and calcite. As it was noted by several authors [21,26], preiswerkite formation is favored by the presence of an aqueous fluid, Si-poor and an Al-Na-rich system. For this reason, preiswerkite-bearing marble petrogenesis is considered further in terms of a metasomatic process, even if it was realized in a local scale.

The common feature of all metasomatic parageneses is a change from less hydrous to more hydrous minerals (amphiboles to micas to chlorite). The reason for the production of such a time sequence of mineral distribution could only be a systematic change in composition and properties of the mineral-forming fluids as they permeate the protolith and react with it.

In the model system K-Na-Mg-Al-Si-$CO_2$-$H_2O$, the most common original rocks (evaporites, clays, etc.) and the secondary minerals developed after them can be described satisfactorily. However, the metamorphism of siliceous dolomites requires an analysis of phase equilibria in the presence of a mixed $H_2O$-$CO_2$ fluid [40]. Let us consider the connection of chemical potentials of $CO_2$ and $H_2O$ in a binary fluid ($CO_2 + H_2O$) at $p_{CO_2} + p_{H_2O} = p_{total} = Const$ and $T = Const$. Under these conditions, the molar fraction of $CO_2$ in the fluid $(x_{CO_2})$ is defined by the expression:

$$x_{CO_2} = \frac{p_{CO_2}}{p_{CO_2} + p_{H_2O}} = \frac{p_{CO_2}}{p_{total}},$$ (6)

where from

$$p_{H_2O} = (1 - x_{CO_2})p_{total}$$ (7)

The values of the chemical potential of $H_2O$ in the fluid $(\mu_{H_2O})$ are determined, as is known, by the expression:

$$\mu_{H_2O} = \mu_{H_2O}^0 + RT \ln f_{H_2O} = \mu_{H_2O}^0 + RT \ln \gamma_{H_2O} p_{H_2O} = \mu_{H_2O}^0 + RT \ln \gamma_{H_2O}(1 - x_{CO_2})p_{total}$$ (8)

where $\mu_{H_2O}^0$ is the standard chemical potential of water at a given temperature and pressure; $f_{H_2O}$, $\gamma_{H_2O}$, $p_{H_2O}$ is fugacity, fugacity coefficient and partial pressure of $H_2O$ in the fluid, respectively; $x_{CO_2}$ is the molar fraction of $CO_2$ in the fluid.

To ascertain the main parameters of the mineral-forming environment, which govern the stability of minerals in the marbles, let us consider the topology of the diagrams plotted in coordinates of the chemical potentials [41]. The analysis of the model system is based on the fundamental principles of Korzhinsky [41,42]: local equilibria exists among minerals, all components have differential mobility and the upcoming fluids react with the protolith. Such an approach has been used successfully to model the chemical transformation of metasomatic rocks [42].

Depending on the mineral and chemical composition of the protolith, two specific versions of preiswerkite-bearing mineral associations can be formed: (1) with minerals of mica group; (2) with minerals of amphibole group.

### 5.2.1. Mineral Equilibria with (Micas)

In plotting the qualitative $\mu_{K^+} - \mu_{Na^+}$ and $\mu_{Na^+} - \mu_{H_2O}$ diagrams of the Luc Yen mica transformations, it was assumed that the mineral composition of the preiswerkite-bearing marbles is governed by four minerals: phlogopite, aspidolite, preiswerkite and corundum. The chemical compositions of the micas are determined by the proportions of Mg, Al and Si. However, in all micas the ratio Mg/Si is equal to one because of substitution Mg[Si]←Al[Al] (see Figure 1). Thus, we obtain two virtually inert components (Mg and Al) (Figure 5).

Thus, the number of minerals (phases) in the model system is taken as four and the number of inert components as two, leaving three completely mobile components (K, Na and $H_2O$). We take the temperature and pressure as constant external equilibrium factors. The equations of the chemical reactions for the lines of monovariant equilibrium were calculated for the theoretical (normative) compositions of the minerals (Table 3), the

chemistry of which is very close to that of their natural analogs. The equations of the chemical reactions are given in Table 4.

**Table 3.** Theoretical compositions of minerals (solid-solution components) adopted for calculating chemical reaction equations.

| Mineral | Symbol | Crystallochemical Formula |
|---|---|---|
| Phlogopite | Phl | $KMg_3(AlSi_3O_{10})(OH)_2$ |
| Aspidolite | Asp | $NaMg_3(AlSi_3O_{10})(OH)_2$ |
| Preiswerkite | Pwk | $Na(Mg_2Al)(Al_2Si_2O_{10})(OH)_2$ |
| Corundum | Crn | $Al_2O_3$ |

**Table 4.** Equations of chemical reactions occurring on lines of monovariant equilibrium.

| N | Lines * | Chemical Reactions |
|---|---|---|
| 1 | [Pwk] | $Phl + Na^+ = Asp + K^+$ |
| 2 | [Crd] | $Phl + Na^+ = Asp + K^+$ |
| 4 | [Phl] | $4Asp + 7Crn + 2Na^+ + 3H_2O = 6Prw + 2H^+$ |
| 3 | [Asp] | $4Phl + 7Crn + 6Na^+ + 3H_2O = 6Prw + 2H^+ + 4K^+$ |

* Reactions are labelled with the phase absent convention ([Pwk] = "Pwk"-absent reaction).

Mineral equilibria in such a system can be depicted graphically in three-dimensional space. Therefore it is more feasible first to analyze in detail the behavior of the lines of monovariant equilibrium of a reaction on the diagram in coordinates of two completely mobile components ($\mu_{K^+}$, $\mu_{Na^+}$), taking the temperature, pressure and chemical potential of $H_2O$ as constant external equilibrium factors and then on the basis of this analysis, to consider the diagram in $\mu_{Na^+} - \mu_{H_2O}$ coordinates.

In the qualitative $\mu_{K^+} - \mu_{Na^+}$ diagram, we consider the transformations of phlogopite into aspidolite (singular Reactions 1,2, Figure 5), phlogopite+corundum into preiswerkite (Reaction 4) and aspidolite+corundum into preiswerkite (Reaction 3). It should be noted that with low values of $\mu_{Na^+}$, preiswerkite breaks down into the association aspidolite + corundum according to Reaction 3, Figure 5. As the chemical potential of potassium increases, preiswerkite becomes unstable and is replaced by phlogopite + corundum (Reaction 4), and as $\mu_{Na^+}$ increases, the field of preiswerkite-bearing parageneses is somewhat broadening (Reactions 2, 4, Figure 5).

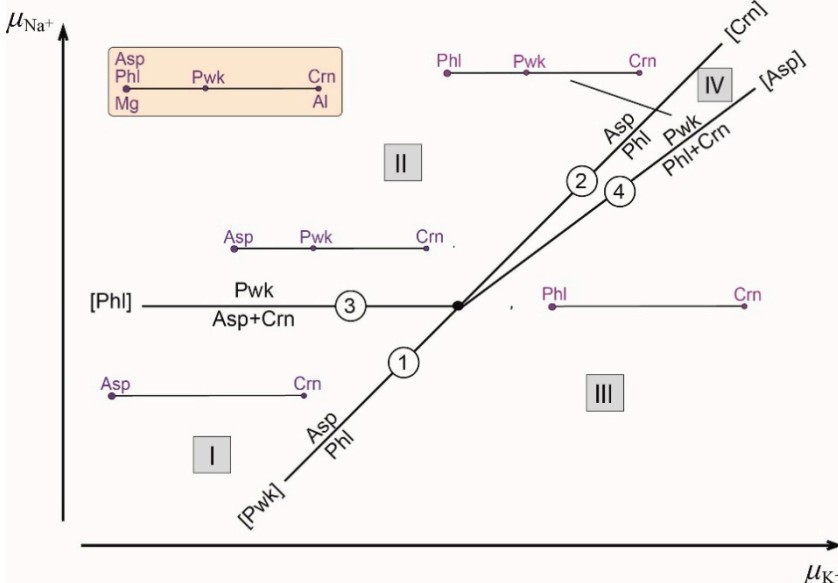

**Figure 5.** Qualitative $\mu_{K^+} - \mu_{Na^+}$ diagram of mica parageneses in the system K-Na-Mg-Al-Si-$H_2O$. I, II, III, IV—stable divariant fields; the numbers in circles indicate the reaction numbers in the Table 4.

In plotting the qualitative $\mu_{Na^+} - \mu_{H_2O}$, we take the temperature, pressure and chemical potential of potassium as constant external equilibrium factors. Mineral equilibria in such a system can be depicted graphically in two-dimensional space (Figure 6).

The formation of preiswerkite in this system is controlled by two reactions (3 and 4, Figure 6), by which it is completely destroyed when $\mu_{H_2O}$ decreases. Under these conditions, the stability of aspidolite (Reactions 1 and 2, Figure 6) is determined only by the chemical potential of sodium in the system and is independent of the $\mu_{H_2O}$ of the mineral-forming medium. It should be noted that preiswerkite is stable in association with phlogopite (field III) at low activities of $Na^+$, and with aspidolite (field IV) at high activities of $Na^+$.

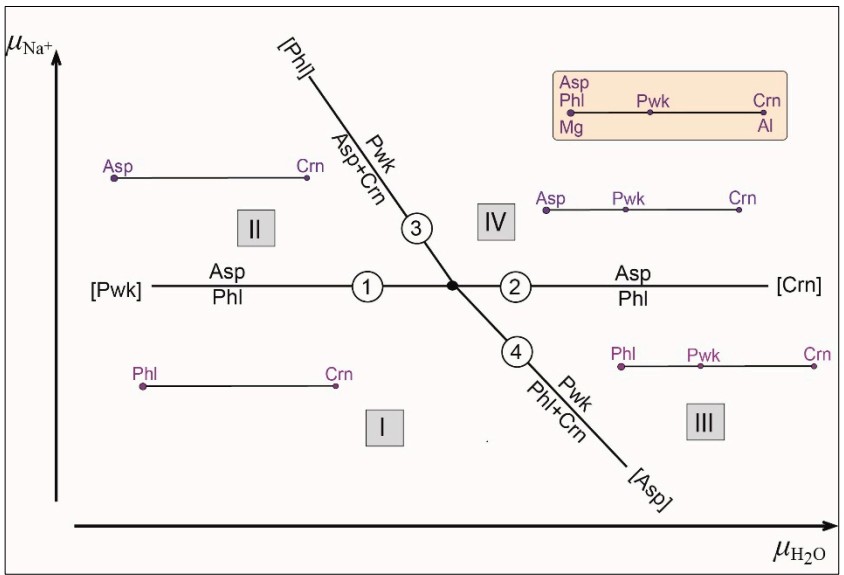

**Figure 6.** Qualitative $\mu_{Na^+} - \mu_{H_2O}$ diagram of mica parageneses in the system Na-Mg-Al-Si-CO$_2$-H$_2$O. I, II, III, IV—stable divariant fields; the numbers in circles indicate the reaction numbers in the Table 4.

### 5.2.2. Mineral Equilibria with Amphibole (Sadanagaite)

In plotting the qualitative diagram ($\mu_{Na^+} - \mu_{H_2O}$) of Na-Mg-Al-Si-CO$_2$-H$_2$O system for the transformations of amphibole into preiswerkite, it assumed that sadanagaite governs the main features of the amphibole composition in the Si-poor and Al-rich pods in marbles of the Luc Yen deposit. Sadanagaite was found here in the associations with preiswerkite, anorthite, dolomite, spinel and corundum. The proportions of the Mg, Al and Si oxides determine the mineral composition of the rocks. Titanium oxide, which is present in the rocks in the form of ilmenite, titanite and rutile, can be considered a separate component. Lumping FeO and MgO together, we obtain three virtually inert components: (Mg, Fe)O, Al$_2$O$_3$, and SiO$_2$. The main components producing the transformation of the original rocks are sodium and H$_2$O.

Thus, the number of minerals (phases) in the model system is taken as six and the number of inert components as three, leaving two completely mobile components. We take the temperature and pressure as constant external equilibrium factors. Mineral equilibria in such a system can be depicted graphically in two-dimensional space. Therefore, we analyze in detail the behavior of the lines of monovariant equilibrium of a reaction on the diagram in coordinates of two completely mobile components ($\mu_{Na^+}, \mu_{H_2O}$).

The equations of the chemical reactions for the lines of monovariant equilibrium were calculated for the theoretical (normative) compositions of the minerals (Table 5), the chemistry of which is close to that of their natural analogs. The equations of the chemical reactions are given in Table 6.

**Table 5.** Theoretical compositions of minerals (solid-solution components) adopted for calculating chemical reaction equations.

| Mineral | Symbol | Crystallochemical Formula |
|---|---|---|
| Preiswerkite | Pwk | $Na(Mg_2Al)(Al_2Si_2O_{10})(OH)_2$ |
| Sadanagaite | Sdg | $NaCa_2(Mg_4Al)(Al_2Si_6O_{22})(OH)_2$ |
| Anorthite | An | $Ca(Al_2Si_2O_8)_2$ |
| Dolomite | Dol | $CaMg(CO_3)_2$ |
| Spinel | Spl | $MgAl_2O_4$ |
| Corundum | Crn | $Al_2O_3$ |

**Table 6.** Equations of chemical reactions occurring on lines of monovariant equilibrium.

| N | Lines * | Chemical Reactions |
|---|---|---|
| 1 | [AnSdg] | $Dol + Crn = Spl + (CaCO_3) + CO_2$ |
| 2 | [AnPwk] | $Dol + Crn = Spl + (CaCO_3) + CO_2$ |
| 3 | [SdgPwk] | $Dol + Crn = Spl + (CaCO_3) + CO_2$ |
| 4 | [SplAn] | $4Sdg + 5Crn + 8Dol + 6Na^+ + 9H_2O = 10Pwk + 16Cal + 6H^+$ |
| 5 | [AnCrn] | $4Sdg + 5Spl + 3Dol + 6Na^+ + 4H_2O + 4H^+ = 10Pwk + 6Cal + 5Ca^{2+}$ |
| 6 | [SplPwk] | $5An + 6Dol + 2Na^+ + 3H_2O = 2Sdg + 7CaCO_3 + 5CO_2 + 2H^+$ |
| 7 | [CrnPwk] | $5An + 6Dol + 2Na^+ + 3H_2O = 2Sdg + 7CaCO_3 + 5CO_2 + 2H^+$ |
| 8 | [SplCrn] | $5An + 6Dol + 2Na^+ + 3H_2O = 2Sdg + 7CaCO_3 + 5CO_2 + 2H^+$ |
| 9 | [SplDol] | $4Sdg + 3Crn + 2Na^+ + 6H^+ = 6Pwk + 4An + H_2O + 4Ca^{2+}$ |
| 10 | [SplSdg] | $2Pwk + 6CaCO_3 + 2CO_2 + 2H^+ = 2An + 4Dol + Crn + 2Na^+ + 3H_2O$ |
| 11 | [AnDol] | $4Sdg + 8Spl + 6Na^+ + H_2O + 10H^+ = 10Pwk + 3Crn + 8Ca^{2+}$ |
| 12 | [SdgDol] | $4Spl + 2An + 2Na^+ + H_2O + 2H^+ = 2Pwk + 3Crn + 2Ca^{2+}$ |
| 13 | [SdgCrn] | $2Prw + 5CaCO_3 + CO_2 + 2H^+ = 2An + Spl + 3Dol + 2Na^+ + 2H_2O$ |
| 15 | [CrnDol] | $4Pwk + An + 4Ca^{2+} + 2H_2O = 2Spl + 2Sdg + 2Na^+ + 8H^+$ |
| 15 | [PwkDol] | $6Spl + 5An + 2Na^+ = 2Sdg + 6Crn + Ca^{2+}$ |

* Reactions are labelled with the phase absent convention ([AnSdg] = "AnSdg"-absent reaction). Reactions 8–15 are metastable in this system (see text).

Thus, let us consider the diagram of the multisystem in $\mu_{Na^+} - \mu_{H_2O}$ coordinates, because water in the mineral-forming medium essentially affects the stability of hydroxyl-bearing minerals (sadanagaite and preiswerkite) and also the equilibrium corundum + spinel (Reactions 1–3, Table 6). It should be noted that Reactions 8–15 (Table 6) are omitted because nonvariant points (Sdg), (Crn) and (Spl) are metastable and dolomite is in excess. Figure 7 gives this diagram in its entirety.

Monovariant singular equilibria of the Reactions 1–3 (Table 6) are controlled only by the chemical potential of $CO_2$ in the fluid and do not depend on the activity of sodium in the medium. Thus, as it was shown above, in case of increasing $\mu_{H_2O}$ ($\mu_{CO_2}$ and $x_{CO_2}$ are decreasing), an association of corundum+dolomite is replaced by spinel according to Reactions 1–3. This monovariant line divides the diagram into two parts (Figure 7). In the left part (fields I-III), only corundum is stable, and in the right part (fields IV-VI), spinel is stable with an excess of dolomite; with an increase of aluminum content, corundum + spinel association is possible

It follows from the diagram that when the values of $\mu_{Na^+}$ are increasing, in a solution with low activity of $H_2O$ (fields I-III) corundum (in excess of dolomite) associated with anorthite (field I), then with sadanagaite (field II) and at maximum sodium activities (field III) preiswerkite+corundum+dolomite association is formed (Reaction 4, Table 6).

At higher values of $\mu_{H_2O}$ (fields IV-VI) with an excess in dolomite, spinel appears instead of corundum and with increasing of $\mu_{Na^+}$, the association spinel+anorthite (field I) is replaced by an association spinel+sadanagaite (field II) and then by an association preiswerkite+corundum+dolomite (field III).

The stability of the main minerals in marbles (in the presence of dolomite $\pm$ calcite in all parageneses) as a function of the chemical potentials of sodium, potassium and $H_2O$ in the mineral-forming medium is clearly illustrated by two-dimensional diagrams.

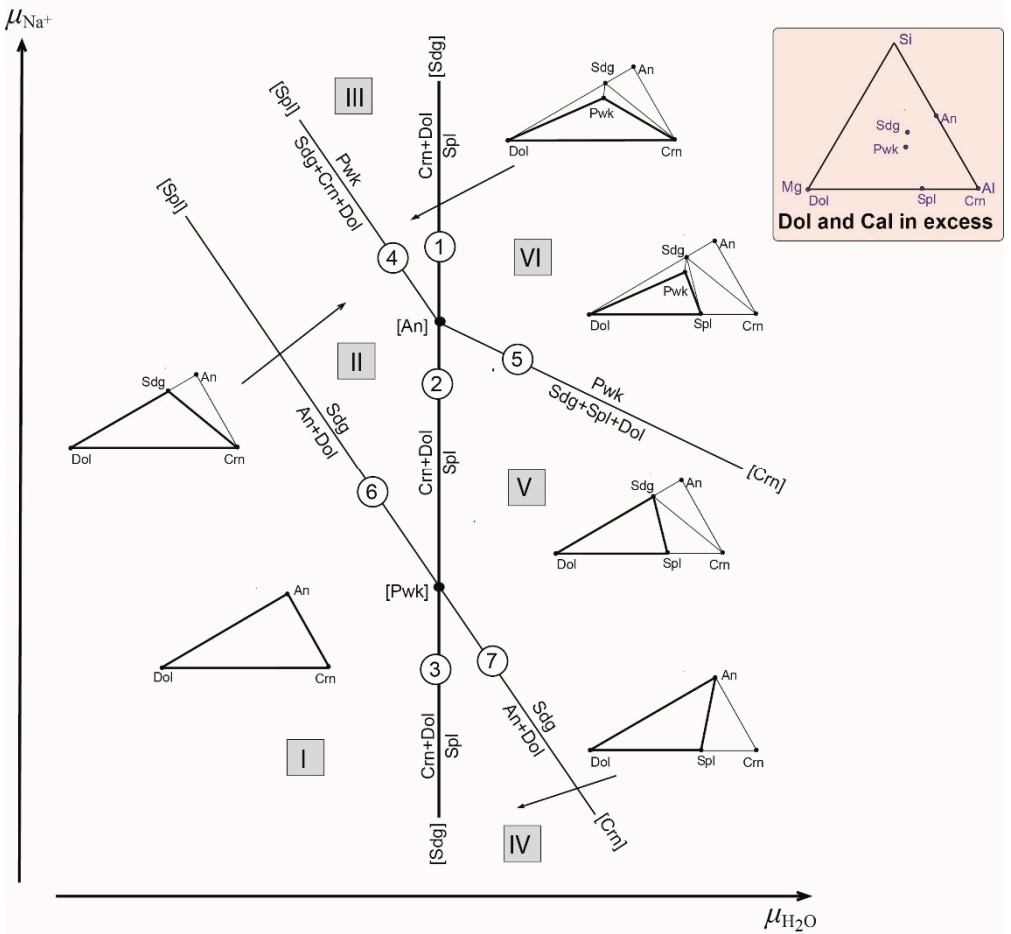

**Figure 7.** Qualitative $\mu_{Na^+} - \mu_{H_2O}$ diagram of mineral parageneses of preiswerkite in the system Na-Mg-Al-Si-CO$_2$-H$_2$O. I, II, III, IV, V, VI—stable divariant fields; the numbers in circles indicate the reaction numbers in the Table 6.

Unfortunately, the obtained diagrams do not allow quantitative analysis of the specific metamorphic conditions that facilitate Na-trioctahedral micas transformations due to the lack of thermodynamic data for these minerals.

## 6. Conclusions

The first occurrence of preiswerkite in pods of marble-hosted gem spinel deposits of the Luc Yen district, northern Vietnam is reported. In marbles it associates, besides calcite and dolomite, also with spinel, sadanagaite and corundum. The average composition of preiswerkite is $(Na_{0.88}Ca_{0.08}K_{0.01})_{\Sigma0.97}(Mg_{2.29}{}^{VI}Al_{0.72}Fe_{0.04})_{\Sigma3.05}[(^{IV}Al_{1.95}Si_{2.05})_{\Sigma4.00}O_{10}](OH)_2$.

A characteristic geochemical feature of the studied preiswerkite is a reduced iron content, which reaches 0.09 atoms per formula unit (1.53 wt.% FeO).

Thermodynamic calculations confirm that the rarity of sodium mica in nature is mainly due to their crystallographic configuration. The detailed study of paragenetic associations revealed two specific versions of preiswerkite-bearing assemblages: (1) with mica group minerals; (2) with amphibole group minerals.

The rarity of preiswerkite in marbles seems to be due to unusual chemical compositions of protolith: it appears in an aqueous fluid Na-Al-rich Si-poor composition principally at the time of marble formation.

Preiswerkite formation in type 3 spinel-bearing marbles implies that local conditions favored its formation in the Luc Yen gem spinel deposit at P-T conditions of amphibolite and granulite facies.

**Author Contributions:** Conceptualization, V.G.K. and P.B.S.; methodology, V.G.K., P.B.S. and K.A.K.; validation, V.G.K. and P.B.S.; investigation, K.A.K., O.Y.M., A.A.Z., V.N.B., T.F.S., G.A.G. and M.E.K.; writing—original draft preparation, V.G.K., K.A.K. and O.Y.M.; writing—review and editing, V.G.K., K.A.K. and P.B.S.; visualization, V.G.K., K.A.K. and O.Y.M.; supervision, V.G.K. and P.B.S.; funding acquisition, P.B.S. and V.G.K. All authors have read and agreed to the published version of the manuscript.

**Funding:** The studies were supported by the Russian Science Foundation, grant no. 22-27-00172.

**Acknowledgments:** We are grateful to Robert Martin and an anonymous reviewer for their constructive and insightful reviews of the first version of the manuscript. The studies in this work were carried out using the equipment of the X-ray Diffraction Studies and Geomodel Resource Centers of St. Petersburg State University, respectively.

**Conflicts of Interest:** The authors declare no conflict of interest.

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
