# Peer review of "Preiswerkite: A First Occurrence in Marble Hosting Gem Spinel Deposits, Luc Yen, Vietnam"

_minerals, doi:10.3390/min12081024_

Round 1
Reviewer 1 Report
Your article on preiswerkite is a welcome addition to the literature on this rare mica. The contribution consists of solid new data on the mica, and a long discourse on the thermodynamic aspects of the occurrence. This discourse is not based on solid thermodynamic parameters, and I view it as a very preliminary assessment. The authors have chosen not to wait for real thermodynamic measurements on aspidolite and preiswerkite to forge ahead with a model. I hope that readers realize that the model is very preliminary. After reading the article, one does not have an evaluation of the conditions of formation of preiswerkite at Luc Yen.
Phlogopite is shown to be replaced by preiswerkite, but no details are offered to document its replacement by aspidolite. Intermediate compositions should be plotted (i.e., shown graphically).
The article seems to have been written in great haste. How do I know this??? 1) There is a mixup in numbering the figures, also reflected in the text. 2) Important mistakes are made in the chemical formulas of eastonite and preiswerkite shown in Figure 1.
Here are specific comments:
Line 2 Preiswerkite: a first occurrence in marble hosting gem spinel deposits, Luc Yen, Vietnam. In my opinion, that title says it all.
Line 3 North Vietnam versus South Vietnam??? I believe that the country’s name is Vietnam
Line 6 Tatyana F. + a space
Lines 7, 13, 436 Is it St. Petersburg State University, Saint-Petersburg State University or Saint Petersburg State University? Also line 112, 518. Decide.
Line 33 northern Vietnam is fine
Line 53 You must write the formulae correctly!
Please use this figure to plot the compositions that are intermediate between phlogopite and preiswerkite. You mention them in the abstract and text, but don’t show them...
Line 67 You are saying that in the other half, Na-Al-poor and Si-rich conditions prevailed in a carbonic fluid? I would remove this statement because the reader will not be in a position to understand the alternative that you have in mind.
Line 74 These minerals are typical of granulite-grade marble.
Line 76 The adjective symplectitic is an appropriate modifier of the word texture, not of the word marble (unless you tell the reader what the symplectitic texture involves in that specimen).
Line 101 The scale bars are illegible. Show them clearly within the area of the photo.
Line 107 Please clarify.
Lines 127-128 You can omit.
Line 158 Rephrase this sentence.
Line 187 96.8% of what population??? not clear
Line 189 94.7% of what population??? not clear
Line 201 In my opinion, sodic micas are inherently less likely to form because of the relatively small ionic radius of the interlayer cation, and the stronger repulsion due to the closeness of the juxtaposed sheets of tetrahedra that results. As a corollary statement, eastonite is more likely to form than preiswerkite. This explanation can replace lines 197 to 201 and perhaps more.
Line 235 often means many times; you mean commonly?
Line 259 Saturation in H2O is a concept that applies to a melt. You do not consider the possibility that a melt is present, and instead attribute the replacement to an aqueous fluid. In my opinion, you cannot refer to saturation in that situation.
Line 400 In my opinion, the Conclusions section is too long. I recommend three or four one-sentence statements that really summarize the contents of the article.
Lines 424-425 This seems to me to be a statement of the obvious. But we are not closer to an estimate of the P and T of the stability field of preiswerkite as a result of this investigation.
Lines 475, 511 spacing between words
Revised abstract
We report a new occurrence of preiswerkite, the rare sodium analog of the trioctahedral mica, from marble-hosted noble spinel deposits of the Luc Yen district, Vietnam. It is found in marble for the first time. The preiswerkite is anhedral and associated with phlogopite, aspidolite, sadanagaite, pargasite, spinel, corundum, dolomite and calcite. Its average composition is (Na0.88Ca0.08K0.01)Σ0.97(Mg2.29VIAl0.72Fe0.04)Σ3.05[(IVAl1.95Si2.05)Σ4.00O10](OH)2. The compositions of preiswerkite have a narrow range of Mg# values (0.96–0.99) and define a preiswerkite–aspidolite solid-solution series. Compared to other occurrences, the Luc Yen preiswerkite has a low iron content, which attains 0.09 atoms per formula unit (1.53 wt. % FeO). The formation of preiswerkite is favored by the proportion of Mg, Al and Si in the precursor limestone and the increased activity of sodium and H2O in the fluid phase.

Author Response
Dear Reviewer,
Many thank you for your comments they were very useful for us.
1) Title of the article slightly changed.
2) We have changed the text according to your comments (see text).
See attached pdf file

Reviewer 2 Report
The submitted manuscript is a very interesting work with a high level of content. It is evident that the authors are specialists in mineralogy. I added some minor comments in the attached file.

Author Response
Dear Reviewer,
Many thank you for your comments they were very useful for us.
We have changed the text according to your comments.
See attached file

Round 2
Reviewer 1 Report
I am satisfied that my suggestions for improvement have been addressed.